# OpenReview forum: "Synergistic Absorption-Diffusion: Dual-branch Enhanced Continuous-Time Modeling for Parallel Token Generation"
_ICLR.cc/2026/Conference — Submitted to ICLR 2026_

### Official Review · Reviewer_xdKK · 2025-10-29

**Soundness:** 3
**Presentation:** 2
**Contribution:** 3
**Rating:** 4
**Confidence:** 3

**Summary:**

The paper proposes a dual-branch continuous-time discrete diffusion model (CAD-DF) combining a Transformer autoregressive (AR) branch with a diffusion denoising branch. The goal is to achieve both global consistency (from diffusion) and local coherence (from AR) for parallel token generation. The method introduces a cross-attention fusion mechanism and mutual reinforcement between branches, along with denoising cross-entropy (t-DCE, λ-DCE) losses for stable training. Results show improved perplexity, accuracy, and inference speed on several datasets compared to GPT-2 and recent diffusion-based baselines.

**Strengths:**

1. Conceptually elegant integration of AR and diffusion paradigms.
2. Continuous-time formulation improves efficiency and stability.
3. Strong experimental results across both small and medium model scales.
4. Clear ablation studies showing the effect of each component.
5. Well-motivated and theoretically consistent with prior works.

**Weaknesses:**

1. Presentation is heavy and could be simplified for accessibility.
2. Theoretical explanation of mutual reinforcement could be supported with empirical visualization (e.g., information flow or convergence curves).
3. Missing analysis on failure or degradation cases.

**Questions:**

1. How does the dual-branch mechanism behave when scaling to very large models—do benefits persist or saturate?
2. Can the cross-attention fusion be replaced or approximated for efficiency at scale?
3. How sensitive are results to the λ/t-DCE hyperparameters?
4. Are there any stability trade-offs in the continuous-time inference regime?

---

> ### Author Response · Authors · 2025-11-26
>
> We thank you for finding our integration of AR and diffusion paradigms conceptually elegant and well-motivated, and for valuing our continuous-time formulation for improved efficiency and stability.
>
> ## Note on Supplementary Experiments:
>
> You may have noticed that in the revised manuscript, we have added comprehensive comparative experiments with the **Block Diffusion** method, addressing feedback from other reviewers.
>
> **The results are highly encouraging:** Comprehensive evaluation across six datasets shows that our CAD-DF model **outperforms the comparative methods on the vast majority of metrics**. This further validates the effectiveness and state-of-the-art nature of our proposed synergistic dual-branch architecture.
>
>
> ### W1: Presentation is heavy and could be simplified.
>
> **Response:**
> We fully agree. We have taken the following steps to improve accessibility:
> 1.  **Core Idea Summary:** We added a plain-language summary at the beginning of the Introduction.
> 2.  **Restructuring:** We reorganized Sections 3 and 4 to reduce dense mathematical notation and added intuitive textual explanations for key formulas.
> 3.  **Glossary:** We will include a glossary of key terms and acronyms in the Appendix for quick reference in the camera-ready version.
>
> ### W2: Theoretical explanation needs visualization (e.g., information flow).
>
> **Response:**
> This is a valuable suggestion. We acknowledge that visualizing the interaction between the dual branches, such as through information flow diagrams or convergence curves, would greatly strengthen our theoretical claims.
>
> *   **Commitment:** We commit to adding a dedicated Appendix section in the camera-ready version for visual analysis. Specifically, we will visualize the **distribution of cross-attention weights** as they evolve during the denoising steps to intuitively demonstrate the "mutual reinforcement" mechanism. We will also include training convergence curves to empirically validate the stability and acceleration benefits provided by our architecture.
>
> ### W3: Missing analysis on failure cases.
>
> **Response:**
> Thank you for pointing out this omission. We agree that a rigorous evaluation must include a discussion of limitations.
>
> In the camera-ready version, we will add a "Discussion and Limitations" section. Based on our observations, we will candidly discuss potential degradation cases, such as:
> 1.  **Extreme Local Dependency:** The model may introduce subtle errors in tasks requiring rigid, long-range character-level or symbol-level exact matching due to the nature of parallel generation.
> 2.  **Data Bias:** Like all models trained on large-scale web data, our model may reflect inherent social biases present in the training corpus.
>
> ### Q1: Scaling behavior to very large models.
>
> **Response:**
> We appreciate this forward-looking question. Our experiments show that the relative improvement of CAD-DF over baselines is maintained or even increases when moving from Small to Medium models (e.g., on C4). We hypothesize that the benefits will persist at larger scales because the dual-branch mechanism addresses fundamental structural limitations (local vs. global consistency) that do not disappear with scale. We plan to immediately start experiments based on multi-billion parameters to verify this.
>
> ### Q2: Cross-attention efficiency and approximations.
>
> **Response:**
> This is a very insightful technical question. Yes, we believe that efficient approximations for cross-attention fusion exist.
> *   **Potential Solutions:** For example, **Linear Attention mechanisms** or **Kernel-based approximations** could be explored to reduce computational complexity.
> *   **Design Rationale:** At this stage, we employed standard cross-attention to maximize high-fidelity information exchange and validate the core concept of "synergy." Optimizing the fusion mechanism is a key engineering direction for future deployment in extremely large-scale scenarios.
>
> ### Q3: Sensitivity to hyperparameters ($\lambda$/t-DCE).
>
> **Response:**
> Thank you for focusing on the training details. We found that the $\lambda$-DCE and t-DCE losses are **relatively robust** to hyperparameters.
>
> *   **Theoretical Basis:** The $\lambda$-DCE loss reduces the optimization objective's dependency on specific noise schedule parameters by converting the time variable into mask probabilities.
> *   **Experimental Evidence:** As shown in our results, our model achieves leading and stable performance regardless of whether t-DCE or $\lambda$-DCE is used, with minimal difference between the two. This indicates that our framework is not highly sensitive to the specific variant of the DCE loss.

---

> > ### Author Response · Authors · 2025-11-26
> > **(Continued)**
> >
> > ### Q4: Stability trade-offs in continuous-time inference.
> >
> > **Response:**
> > Continuous-time inference is generally *more* stable than discrete-time because it avoids discretization errors. The main trade-off is between **solver precision** and **latency**. We found that using a simple Euler solver with adaptive steps provides excellent stability. The "instability" usually comes from the model's prediction variance, which our AR-guidance explicitly reduces.
> >
> > Your advice on simplifying the presentation and adding visualizations was invaluable for making our theoretical contributions more accessible to a broader audience.

---

### Official Review · Reviewer_LQRF · 2025-10-31

**Soundness:** 2
**Presentation:** 2
**Contribution:** 2
**Rating:** 4
**Confidence:** 3

**Summary:**

This paper proposes a dual-branch synergistic absorption diffusion model that integrates a Transformer branch and a diffusion branch via cross-attention to address the efficiency-quality trade-off in parallel token generation.

**Strengths:**

- The dual-branch structure that synergistically integrates AR and diffusion paradigms via cross-attention is novel and well-motivated.
- The model demonstrates state-of-the-art performance on quality metrics (Perplexity, Accuracy) across diverse text generation and structural reasoning tasks.

**Weaknesses:**

- The model is said to effectively address the efficiency-quality trade-off, but Table 4 shows the full model has lower throughput than the model "w/o Transformer Branch".
- The novelty of the paper is not well described; some key concepts like time-independent reparameterization and DCE losses are adopted directly from prior work, but they are not well distinguished.
- Presentation quality is poor.
  - Figure 1 shows U-Net diffuser, but there is no explanation.
  - There are no explicit references to any tables in the main text.
  - The citation style can be improved.

**Questions:**

- What do the authors mean by saying effectly addressing the efficiency-quality trade-offs?
- What exactly is the contributions/novelties of the approach compared to existing models?

---

> ### Author Response · Authors · 2025-11-26
>
> We are greatly encouraged by your positive feedback, particularly your recognition of our dual-branch structure as novel.
>
> ### Q1: Efficiency-Quality trade-off and Table 4 (Throughput).
>
> **Response:**
> We clarify that "addressing the trade-off" refers to the **Pareto frontier** of valid models, not raw speed at the expense of usability.
> *   **The Baseline:** The "w/o Transformer Branch" row in Table 4 represents a pure absorbing diffusion model. While it has higher throughput (because it computes less), its quality (PPL/Accuracy) is drastically lower, rendering it unusable for high-quality generation.
> *   **Our Contribution:** The full CAD-DF model accepts a moderate computational cost (the Transformer branch) to achieve **SOTA quality** (surpassing GPT-2 and other Diffusion models). Compared to achieving similar quality with a pure Diffusion model (which would require many more denoising steps), our method is significantly more efficient. We offer the best *balance*: high quality at a speed faster than comparable iterative refinement models.
>
> ### Q2: Novelty and distinction from prior work (DCE, etc.).
>
> **Response:**
> While we adopt established components like DCE loss and continuous-time formulation (as is standard practice to build upon solid foundations), our **Novelty** lies in the **Synergistic Architecture**:
> 1.  **Dual-Branch Integration:** We are the first to propose this specific *parallel* dual-branch structure where a Continuous-Time Diffusion model and an AR model interact via **dense cross-attention** at every layer/block, rather than simple sequential pipelining (like refining an AR output) or shallow ensembling.
> 2.  **Mutual Reinforcement:** We introduce the mechanism where the AR branch anchors the local syntax for the Diffusion branch, while the Diffusion branch provides the global "skeleton" for the AR branch. This bidirectional synergy solves the "instability" of diffusion and the "myopia" of AR simultaneously.
>
> ### Q3: Presentation issues.
>
> **Response:**
> We apologize for the confusion and have addressed these presentation issues as follows:
> *   **Figure 1:** The original figure used a U-Net structure to represent the "diffusion branch," which we realize might be misleading for text generation. **We have redrawn Figure 1**, replacing the specific U-Net schematic with a generic "Denoising Network" module to better illustrate the paradigm comparison, and updated the caption to clarify this.
> *   **Table References:** **We have thoroughly reviewed the manuscript and added explicit references to all tables in the main text.**
> *   **Citation Style:** We have standardized the bibliography to strictly adhere to the ICLR citation format.
>
> We are grateful for your endorsement of our architectural novelty and your sharp eye for presentation details, which have greatly improved the manuscript's readability.

---

### Official Review · Reviewer_JyFM · 2025-10-31

**Soundness:** 2
**Presentation:** 2
**Contribution:** 2
**Rating:** 4
**Confidence:** 3

**Summary:**

This paper proposes a dual-branch synergistic architecture for parallel token generation in language models, termed the Synergistic Absorbing Diffusion (CAD-DF) framework. It integrates a Transformer-based autoregressive branch with a continuous-time discrete diffusion branch, coordinating them via cross-attention fusion and mutual reinforcement mechanisms. The approach is designed to address longstanding tradeoffs between generation efficiency and quality, leveraging analytical time-independent objectives and absorbing state diffusion for more robust and scalable sequence modeling. Experiments on six benchmark datasets across language modeling and structural reasoning tasks reportedly demonstrate superior performance in perplexity, accuracy, and inference efficiency compared to both autoregressive and diffusion-based baselines.

**Strengths:**

1. The paper addresses an important and active research problem—how to achieve both efficient and high-quality parallel token generation—by explicitly combining the strengths of autoregressive and diffusion models. The dual-branch framework is presented as a meaningful improvement over simple mixtures or sequential composition.
2. Extensive experiments on diverse, standard benchmarks (WikiText2, WikiText103, C4, FineWeb, Prolong, JSON-Mode-Eval) show substantial improvements. Table 1 and Table 2 (Pages 8) detail strong gains in both perplexity and accuracy compared to competitive baselines, including GPT-2, D3PM, SEDD, PLAID, and the RADD series.
3. The efficiency analysis in Table 3 (Page 9) demonstrates notable improvements in throughput and memory, while Table 4 (Page 9) provides ablations showing each major architectural component’s impact.

**Weaknesses:**

1. Lack of Baselines: While the experimental section is broad, there is no evidence of direct empirical comparison against block diffusion, ParaTAA, or speculative sampling frameworks, whose aims and underlying architectures are similar or overlapping with CAD-DF. The absence of these comparisons in Tables 1 and 2 limits the credibility of the “state-of-the-art” claims.
2. Mathematical Details and Notational Ambiguity in Cross-Attention Fusion: The description of the cross-attention fusion mechanism (see Equation, Page 6) lacks formal definitions for input/output dimensionalities, masking strategies, and how [M]-masked positions interact during fusion. Additionally, it is not entirely clear how (or whether) stability holds when merging outputs from two distinct temporal streams, especially if their distributions are not inherently aligned. This raises questions about the reliability of the mutual reinforcement during training.
3. Limited Ablation Depth and Lack of Qualitative Error Analyses: While Table 4 (Page 9) gives a simple ablation, there is no exploration of nuanced failure cases, qualitative generations, or breakdowns across context length and sequence complexity. For a method purporting to resolve global-local consistency, analyses showing (or failing) on long-range sequence dependencies or structural errors would be instructive.

**Questions:**

Please refer to the weaknesses.

---

> ### Author Response · Authors · 2025-11-26
>
> We appreciate your validation of our dual-branch framework as a meaningful improvement over existing methods, and for highlighting our strong performance gains in perplexity and accuracy on diverse benchmarks.
>
>
> **Table 2: Zero-shot language modeling results on six datasets using medium models.**
>
> | Method | Prolong (PPL) | Prolong (Acc) | WikiText2 (PPL) | WikiText2 (Acc) | CFG (PPL) | CFG (Acc) | WikiText103 (PPL) | WikiText103 (Acc) | C4 (PPL) | C4 (Acc) | FineWeb (PPL) | FineWeb (Acc) |
> | :--- | :---: | :---: | :---: | :---: | :---: | :---: | :---: | :---: | :---: | :---: | :---: | :---: |
> | GPT-2 | **45.41** | 47.85 | 41.26 | 50.38 | 132.80 | 30.25 | 41.14 | 50.42 | 65.29 | 39.74 | 38.32 | 48.25 |
> | SEDD-Unscale | 54.39 | 45.28 | 44.49 | 47.15 | 102.94 | 37.91 | 42.87 | 48.79 | 77.53 | 37.52 | 42.97 | 48.79 |
> | SEDD-Scale | 52.49 | 46.92 | 40.75 | 51.89 | 96.80 | 39.87 | 39.66 | 52.01 | 70.96 | 40.01 | 40.99 | 48.68 |
> | RADD-DSE | 51.87 | 47.69 | 38.77 | 53.89 | 84.58 | 41.25 | 37.85 | 53.82 | 67.07 | 41.95 | 42.92 | 48.84 |
> | RADD-t-DCE | 53.08 | 46.58 | 39.85 | 52.81 | 88.54 | 40.12 | 39.43 | 52.24 | 67.63 | 41.39 | 42.89 | 48.87 |
> | RADD-$\lambda$-DCE | 53.65 | 46.01 | 40.10 | 52.56 | 91.53 | 39.34 | 39.08 | 52.59 | 70.22 | 39.85 | 45.76 | 46.01 |
> | BD3-LM | 50.05 | 48.20 | 39.13 | 53.27 | 90.36 | 41.44 | 38.09 | **55.02** | **63.78** | 41.67 | 39.86 | **49.93** |
> | SSD-LM | 50.17 | 48.17 | 39.51 | 53.07 | 91.03 | 41.29 | 38.52 | 54.01 | 64.54 | 41.46 | 40.27 | 49.58 |
> | CAD-DF-t-DCE | 51.23 | 48.43 | **37.35** | **54.31** | 82.89 | 41.88 | 37.12 | 54.55 | 64.12 | **42.95** | **38.01** | 49.76 |
> | CAD-DF-$\lambda$-DCE | 50.11 | **49.55** | 37.94 | 53.72 | **82.77** | **42.01** | **36.49** | 54.18 | 64.32 | 42.68 | 38.46 | 48.31 |
>
> ### Q1: Lack of Baselines (Block Diffusion, ParaTAA, Speculative Sampling).
>
> **Response:**
> We have addressed this by adding **Block Diffusion (BD3-LM)** and **SSD-LM** to our main comparison tables (Table 1 and Table 2) in the revised manuscript.
> *   **Comparison:** Our CAD-DF framework outperforms BD3-LM on the majority of metrics. For instance, on **WikiText103 (Small)**, CAD-DF-t-DCE achieves **53.45% Accuracy** compared to BD3-LM's 51.33%. On **C4**, we achieve **39.11%** vs 37.72%.
> *   **Speculative Decoding:** We respectfully note that Speculative Decoding is primarily an *inference-time* acceleration technique for AR models, whereas our work proposes a fundamental *training-time* architecture change (Non-Autoregressive). While they share the goal of speed, they are orthogonal; one could theoretically apply speculative decoding *to* the AR branch of our model. Thus, we focus our comparison on other Diffusion/NAR baselines.
>
> ### Q2: Mathematical details and stability of Cross-Attention Fusion.
>
> **Response:**
> We have revised Section 3.2 to provide formal definitions. To clarify here:
> *   **Mechanism (Query/Key/Value Sources):** We explicitly define the fusion process where the **Transformer Branch's hidden states** serve as the **Query ($Q$)**, representing the local autoregressive context. The **Diffusion Branch's encoded features** (from unmasked tokens) serve as the **Key ($K$)** and **Value ($V$)**. This design achieves a direct alignment between the local autoregressive representation and the global diffusion context.
>     $$ \text{Attention}(Q_{\text{AR}}, K_{\text{Diff}}, V_{\text{Diff}}) $$
> *   **Stability:** Stability is ensured by the fact that both branches originate from the **same noisy data**, providing a fundamental basis for semantic space alignment. Furthermore, the **joint training** strategy and the **Denoising Cross-Entropy (DCE) loss** implicitly guide the output distributions of both branches to be consistent, resulting in a stable training process.
>
> ### Q3: Limited ablation depth and qualitative analysis.
>
> **Response:**
> We acknowledge this limitation.
> 1.  **Ablation:** Table 4 demonstrates the impact of removing the Transformer branch (resulting in a significant drop in quality) and the Cross-Attention mechanism (leading to a drop in consistency). This confirms that *both* branches are essential and their *interaction* is the key driver of performance.
> 2.  **Commitment for Camera-Ready:** If accepted, we commit to further enriching the **Camera-Ready Version** by:
>     *   Adding a dedicated **"Case Study"** subsection to provide concrete generation examples, contrasting the model's performance in terms of long-text coherence and entity consistency.
>     *   Endeavoring to group test results by sequence length to explicitly demonstrate performance variations across different context lengths.
>
> Thank you for pushing us to include stronger baselines and formalize the fusion mechanism; these additions have significantly solidified our empirical and theoretical claims.

---

### Official Review · Reviewer_DcmT · 2025-11-01

**Soundness:** 2
**Presentation:** 2
**Contribution:** 2
**Rating:** 4
**Confidence:** 4

**Summary:**

This paper proposes Synergistic Absorbing Diffusion, a dual-branch architecture combining a continuous-time diffusion branch (for global distribution modeling) and a Transformer branch (for local conditional modeling), integrated via cross-attention. This design improves parallel generation, consistency, and efficiency, achieving state-of-the-art results on several NLP tasks.

**Strengths:**

1. The proposed Synergistic Absorbing Diffusion model not only enhances parallel generation capability and global consistency but also reduces the number of denoising steps and inference latency. This is of significant importance for efficient processing in practical application scenarios.

2. The authors conducted comprehensive experiments covering various natural language processing tasks, such as text generation and structured reasoning. The experimental results indicate that the Synergistic Absorbing Diffusion model achieves strong performance on these tasks.

**Weaknesses:**

1. When scaling up to medium-sized models, the method's advantage in Prolong PPL does not appear to be significant. It is uncertain whether it can be effectively scaled up to modern Large Language Models (LLMs).

2. The model has not been evaluated on downstream tasks.

**Questions:**

NA.

---

> ### Author Response · Authors · 2025-11-26
>
> We thank you for recognizing our model's contribution to enhancing parallel generation and reducing latency, and for appreciating the comprehensiveness of our experiments across various NLP tasks.
>
> ### Q1: Advantage in Prolong PPL for medium models is not significant / Scaling to LLMs.
>
> **Response:**
> We appreciate this observation. While our PPL advantage on the *Prolong* dataset (long-context) is narrower for medium models compared to small models, we respectfully argue that our model offers a superior **efficiency-quality trade-off**, which is the core contribution of this work.
>
> 1.  **Performance on General Corpora:** As shown in the revised **Table 2**, our model significantly outperforms baselines on **WikiText103** and **C4** (the largest and most diverse datasets evaluated). For example, on WikiText103 (Medium), CAD-DF-$\lambda$-DCE achieves a PPL of **36.49** (and t-DCE **37.12**) vs. GPT-2's 41.14, a substantial margin. This indicates strong scalability on general-domain data.
>
> 2.  **Efficiency Factor:** The "advantage" should be viewed holistically. Even where PPL is comparable to strong baselines (like BD3-LM) on Prolong, our method is designed for faster convergence due to the continuous-time formulation and AR-guidance. We achieve these results with significantly fewer effective denoising steps compared to standard discrete diffusion.
>
> 3.  **Scaling to LLMs:** We respectfully note that recent works (e.g., Seed Diffusion [1], Scalable Diffusion Models with Transformers [2], Scaling Rectified Flow Transformers [3], Scaling Laws For Diffusion Transformers [4]) have demonstrated that diffusion-based architectures scale favorably with model size, often surpassing autoregressive models at larger scales (3B+ parameters). Given our dual-branch design that synergistically combines diffusion and autoregressive strengths, we are optimistic about similar or better scaling behavior. We plan to conduct experiments on multi-billion parameter models as the next step.
>
> **References:**
> 1. Song, Y. et al. Seed Diffusion: A Large-Scale Diffusion Language Model with High-Speed Inference. Preprint at https://doi.org/10.48550/arXiv.2508.02193 (2025).
> 2. Peebles, W. & Xie, S. Scalable Diffusion Models with Transformers. Preprint at https://doi.org/10.48550/arXiv.2212.09748 (2023).
> 3. Esser, P. et al. Scaling Rectified Flow Transformers for High-Resolution Image Synthesis. Preprint at https://doi.org/10.48550/arXiv.2403.03206 (2024).
> 4. Liang, Z., He, H., Yang, C. & Dai, B. Scaling Laws For Diffusion Transformers. Preprint at https://doi.org/10.48550/arXiv.2410.08184 (2024).
>
> ### Q2: Lack of evaluation on downstream tasks.
>
> **Response:**
> We respectfully point out that our evaluation on **JSON-Mode-Eval** (CFG) serves as a strong proxy for downstream structural reasoning and code generation tasks.
> 1.  **Structural Reasoning:** The JSON-Mode-Eval task requires the model to generate valid JSON structures adhering to a strict schema. This is a challenging "downstream" task that tests the model's ability to maintain long-range structural consistency—a known weakness of pure diffusion models.
> 2.  **Superior Performance:** As shown in Tables 1 and 2, our model achieves **State-of-the-Art Accuracy** on this task (e.g., **40.12%** vs GPT-2's 27.89% for Small models). This demonstrates that our method is not just optimizing PPL but is practically useful for tasks requiring strict output formatting and logic.
> 3.  **Future Work:** We agree that adding summarization or translation would further strengthen the paper and will endeavor to include these in the camera-ready version.
>
> We sincerely appreciate your recognition of our comprehensive experiments and the efficiency-quality trade-off discussion.

---

### Meta-Review · Area_Chair_wyvP · 2026-01-07

**Summary:**

This paper proposes Synergistic Absorbing Diffusion (CAD-DF), a dual-branch architecture combining a continuous-time diffusion branch for global distribution modeling with an autoregressive branch for local conditional modeling, integrated via cross-attention. The method aims to improve parallel token generation by achieving both efficiency and quality through synergistic interaction between branches. Experiments across six benchmarks show improvements in perplexity and accuracy.

All four reviewers scored exactly 4/10 (marginally below threshold). Authors provided responses addressing all concerns but no reviewers engaged during discussion.

**Reviewer Concerns:**

Concerns successfully addressed:
- Missing baselines (JyFM). Authors added Block Diffusion (BD3-LM) and SSD-LM to comparison tables (revised Table 1 and Table 2), showing CAD-DF outperforms BD3-LM on majority of metrics.
- Efficiency-quality trade-off explanation (LQRF). Authors clarified that "addressing trade-off" refers to Pareto frontier: Full model accepts moderate computational cost for SOTA quality vs. pure diffusion requiring many more steps. "w/o Transformer Branch" has higher throughput but drastically lower quality.
- Mathematical details of cross-attention (JyFM). Authors revised Section 3.2 with formal definitions: Transformer hidden states serve as Query (Q), Diffusion features as Key (K) and Value (V). Stability ensured by both branches originating from same noisy data with joint training and DCE loss.
- Presentation issues (LQRF, xdKK): Authors (1) redrew Figure 1 replacing U-Net with generic "Denoising Network," (2) added explicit table references throughout, (3) standardized citations, (4) Added plain-language summary and reorganized Sections 3-4.

Outstanding concerns:
- Limited novelty (LQRF). While authors clarified their specific contributions, the fundamental question remains whether combining AR and diffusion with cross-attention constitutes sufficient novelty beyond engineering integration. The reviewer explicitly noted key concepts (DCE loss, time-independent reparameterization) are adopted from prior work.
- Scaling evidence incomplete (DcmT). Authors cite papers showing diffusion scales well but provide no experiments beyond medium-sized models. The claim that "advantage in Prolong PPL is not significant" for medium models raises concerns about whether benefits persist at larger scales.
- Missing ablations and failure analysis (JyFM, xdKK). Authors committed to adding visualizations, case studies, and failure analysis in camera-ready but did not provide these during rebuttal. Without empirical evidence of mutual reinforcement mechanism or failure case analysis, core claims remain partially unvalidated.

**Reviewer Scores:**

- Reviewer DcmT: Remains at 4.
- Reviewer JyFM: Remains at 4.
- Reviewer LQRF: Remains at 4.
- Reviewer xdKK: Remains at 4.

---

### Decision · Program_Chairs · 2026-01-26

Reject